# Myorelaxant Effect of Transdermal Cannabidiol Application in Patients with TMD: A Randomized, Double-Blind Trial

**DOI:** 10.3390/jcm8111886

**Published:** 2019-11-06

**Authors:** Aleksandra Nitecka-Buchta, Anna Nowak-Wachol, Kacper Wachol, Karolina Walczyńska-Dragon, Paweł Olczyk, Olgierd Batoryna, Wojciech Kempa, Stefan Baron

**Affiliations:** 1Department of Temporomandibular Disorders, Medical University of Silesia in Katowice, 2 Traugutta sq, 41-800 Zabrze, Poland; nowak.anna@med.sum.edu.pl (A.N.-W.); kacper.wachol@med.sum.edu.pl (K.W.); karolina.dragon@sum.edu.pl (K.W.-D.); sbaron@sum.edu.pl (S.B.); 2Department of Community Pharmacy, Medical University of Silesia, 41-200 Sosnowiec, Poland; polczyk@sum.edu.pl (P.O.); obatoryna@sum.edu.pl (O.B.); 3Faculty of Applied Mathematics, Silesian University of Technology, 44-100 Gliwice, Poland; wojciech.kempa@polsl.pl

**Keywords:** cannabidiol, CBD, myofascial Pain, TMD, bruxism, EMG, masseter muscle

## Abstract

(1) Background: The healing properties of cannabidiol (CBD) have been known for centuries. In this study, we aimed to evaluate the efficiency of the myorelaxant effect of CBD after the transdermal application in patients with myofascial pain. (2) Methods: The Polish version of the Research Diagnostic Criteria for Temporomandibular Disorders (RDC/TMD Ia and Ib) was used. A total of 60 patients were enrolled in the study and were randomly divided into two groups: Group1 and Group2. The average age in Group1 was 23.2 years (SD) = 1.6 years) and in Group2, it was 22.6 years (SD = 1.86). This was a parallel and double-blind trial. Group1 received CBD formulation, whereas Group2 received placebo formulation for topical use. The masseter muscle activity was measured on days 0 and 14, with surface electromyography (sEMG) (Neurobit Optima 4, Neurobit System, Gdynia, Poland). Pain intensity in VAS (Visual Analogue Scale) was measured on days 0 and 14. (3) Results: in Group1, the sEMG masseter activity significantly decreased (11% in the right and 12.6% in the left masseter muscles). In Group2, the sEMG masseter activity was recorded as 0.23% in the right and 3.3% in the left masseter muscles. Pain intensity in VAS scale was significantly decreased in Group1: 70.2% compared to Group2: 9.81% reduction. Patients were asked to apply formulation twice a day for a period of 14 days. (4) Conclusion: The application of CBD formulation over masseter muscle reduced the activity of masseter muscles and improved the condition of masticatory muscles in patients with myofascial pain.

## 1. Introduction

Nowadays, dental practitioners are more obliged to treat patients suffering from myofascial pain (MFP). The main syndrome, which forces patients to search for medical help, is face and neck pain, headache and pain in the ear. Bruxism, the common sleep disorder and parafunctional activity of the masticatory system, may be considered one of the major causes of tooth wear and masticatory muscles MFP. Bruxism—determined by an increased activity of the limbic system, together with the modification of excitability of the neuromuscular spindles, leading to an increase in muscle contraction strength, which persists despite the end of motor function—constitutes a risk factor for the development of temporomadibular disorders (TMD). Bruxism is a common phenomenon that can affect from 8 to even 31% of the population without significant differences in relation to sex. Depending on the time of occurrence, we can distinguish awake bruxism (AB) and sleep bruxism (SB) [1]. AB is a masticatory muscle activity during wakefulness that is characterized by repetitive or sustained tooth contact and/or by bracing or thrusting of the mandible and is not a movement disorder in otherwise healthy individuals [2]. SB is a masticatory muscle activity during sleep that is characterized as rhythmic (phasic) or non-rhythmic (tonic) and is not a movement disorder or a sleep disorder in otherwise healthy individuals [2]. In addition, bruxism in the waking state is referred to as “centric”, characterized by vertical loading of teeth, while bruxism during sleep is referred to as "eccentric" characterized by horizontal displacement of teeth and grinding during sleep. SB can occur in about 13% of the adult population. In childhood, this adverse phenomenon has the highest frequency, 14–20%, and this decreases with age, reaching 3% among older people [3].

Myofascial pain (MFP) within masseter muscles is a common disorder. Excessive muscle effort in bruxers may develop muscle pain. The main syndrome of myofascial pain is a trigger point: a hard, painful on compression, palpable, localized nodule. Myofascial pain is a symptom of muscle damage. Energy crisis is the reason for the initial sarcomere contracture, that leads to increased metabolism and local hypoxia, muscle damage and inflammatory mediators release. Contraction knots are formed, as an effect of local injury, ischemia and fiber lock. The blood flow around and within the trigger point is reduced. Current approaches for trigger point management are needling, injections and deep massage.

Recently, there has been a heated debate regarding the controversial issue on the legalization of marijuana for medical purposes. The leaves of marijuana along with the blossoms and the fruiting tips of hemp are rich in tetrahydrocannabinol (THC). *Cannabis sativa* L. is known to contain more than 565 chemical compounds that belong to different groups, such as flavonoids, dihydrostilbens, phenanthrenes and the most characteristic for this plant—cannabinoids, mainly psychoactive THC, cannabidiol (CBD), and cannabinol (CBN) [4,5]. The above-mentioned constituents—occurring in the number of 120 active, mainly represented by psychoactive ∆9-tetrahydrocannabinol (THC), cannabidiol (CBD) and cannabinol (CBN)—belong to the C21 or C22 terpenophenolic group of compounds synthesized by the alkylation of resorcinol with a monoterpene unit [6]. The content of individual phytocanabinoids in the plant raw material depends on the growing conditions, geographical location, plant processing methods and the plant variety or chemotype which automatically affects the pharmacological effect of the cannabis extract [7]. Psychoactive effect of THC limits its use in clinical practice.

Despite the multi-character nature, cannabis is widely regarded as constituting only one very diverse species, *C. sativa* L. Within which we can distinguish three varieties: *C. sativa*, *C. indica* and *C. ruderalis* (var. sativa, var. indica, and var. ruderalis, respectively) [7]. Botanical Nomenclature *Cannabis sativa* L. [8]:Kingdom: Plantae (plants)Subkingdom: Tracheobionta (vascular plants)Superdivision: Spermatophyta (seed plants)Division: Magnoliophyta (flowering plants)Class: Magnoliopsida (dicotyledons)Subclass: HamamelididaeOrder: UrticalesFamily: CannabaceaeGenus: CannabisSpecies: sativa

*Cannabis sativa* L. is an annual plant belonging to the order Urticales, family Cannabaceae, and genus Cannabis (hemp) [9]. The highest concentration of THC and CBD is in female cannabis inflorescence [9]. Among the different species of cannabis, its use depends on the application and content of cannabinoids. Industrial varieties are cultivated for seeds in order to obtain oil and hemp fiber (the content of THC is usually less than 1%) [10]. Polyunsaturated fatty acids are an essential component of cannabis, mainly found in the oil obtained from industrial varieties. The polyunsaturated fatty acids are mainly linoleic (omega-6) and linolenic (omega-3) acids, which are in the proportion which is recommended for human dietary intake. In addition, cannabis contains all exogenous, indispensable amino acids [11]. Varieties containing high content of THC are used in medicine. This compound is capable of inhibiting cyclooxygenase enzyme, which blocks prostaglandin synthesis, thereby alleviating the symptoms associated with inflammation [4,12].

CBD is an organic chemical compound belonging to the group of cannabinoids, which is present in hemp [13]. Unlike its isomer, THC, CBD does not have a psychoactive effect, but it affects the psychoactive induction pathway caused by the action of THC [14]. CBD possesses anti-inflammatory and anti-nociceptive properties [15,16,17,18].

According to a publication in 2017 by the National Academies of Sciences, Engineering and Medicine Waschington DC, cannabis might be used to treat chronic pain in adult patients [19]. In a systematic review conducted by the American Academy of Neurology regarding the efficacy and safety of medical marijuana, the authors stated that cannabinoids are effective in reducing spasticity and painful spasm in patients with multiple sclerosis [20]. According to Koppel, cannabinoids are effective at reducing pain from 12–15 weeks up to 1 year of therapy [20].

In the human body, cannabinoids show their effect via two types of cannabinoid receptors: CB_1_—found primarily in the central nervous system and CB_2_—which occurs circumferentially and mainly in the immune system [5]. CB receptors belong to the family of G-protein coupled receptors, physiologically activated by endocannabinoids, arachidonic acid amides and esters such as anandamide and arachidonoylglycerol [21]. Both CB receptor subtypes lead to inhibition of adenylate cyclase and thus to reduced formation of the intracellular messenger substance cyclic adenosine monophosphate (cAMP). CB_1_ receptors are activated by THC and CBD and are responsible for the local analgesic effect [22]. The chemical structure of THC and CBD is slightly different, resulting in a different mechanism of action (Figure 1). THC acts as an agonist of cannabinoid receptors and CBD acts as an antagonist [4]. CBD was shown to enhance the role of THC in neuropathic pain model [23] and in another study, it was found to antagonize the effects of THC [24]. The effect may be dose-dependent [25].

For the first time, cannabinoids are being used in medicine to treat pain [26]. The properties of cannabis that might result in its therapeutic use are mood improvement, sedation, and muscle relaxation [27]. There are already several cannabinoid-based drugs in the world used for therapeutic purposes. Two are currently available in Canada: Nabilone (Cesamet^®^, Valeant Pharmaceuticals Inc., Laval, Canada, CB1 cannabinoid receptor agonist, and Nabiksimole (Sativex^®^, Bayer Schering Pharma, Leverkusen, Germany), an extract containing delta-9-THC and CBD. Another cannabinoid-based drug available in the US is dronabinol (Marinol^®^, Abbott Laboratories, Chicago, IL, USA), a synthetic delta-9-tetrahydrocannabinol. Nabilone is an oral drug approved by Health Canada for the treatment of severe chemotherapy-induced nausea and vomiting. Dronabinol, also in oral form, has also been approved for the treatment of severe chemotherapy-induced nausea and vomiting, as well as weight loss associated with acquired immunodeficiency syndrome. Nabiksimole is in the form of an aerosol for use on the oral mucosa, as an adjunct treatment to spasticity or neuropathic pain in adult patients with multiple sclerosis and incurable cancer pain [28].

The activity of cannabinoids is associated with the modulation of adenylate cyclase and the operation of ion channels—by blocking N-type calcium channels, THC decreases the release of acetylcholine in the hippocampus and norepinephrine from sympathetic nerve terminals [29]. Thus, acetylcholine is not available to bind to the postsynaptic receptors thereby causing muscle contraction. This mechanism of action is interesting to various groups of researchers. The potential myorelaxant activity of CBD has become the subject of this study.

## 2. Experimental Section

This is a parallel-group, double-arm, randomized, double-blind clinical trial conducted to assess the myorelaxantand antinociceptive effect of CBD. A total of 60 patients were enrolled in this study and were provided with CBD formulation for dermal application. The enrolled patients were referred to the Department of Temporomandibular Disorders at the Medical University of Silesia in Katowice, Poland.

### 2.1. Study Participants

In this study, patients were randomly selected from a group of 87 patients attending the Department of Temporomandibular Disorders, Medical University of Silesia in Zabrze, Poland. Patients (*n* = 60) were randomly divided into two groups by allowing them to pick up a number (even or odd) from the container with formulation (Figure 2). The groups were structured as follows: experimental group (Group1, *n* = 30, which includes 18 females and 12 males) and control group (Group2, *n* = 30, which includes 15 females and 15 males) (Figure 3). The average age in Group1 was 23.2 years (SD = 1.6) and in Group2 was 22.6 years (SD = 1.86) (Table 1). Patients were subjected to a preliminary qualification test, which included dental examination, functional assessment of the stomatognathic system, subjective anamnesis, and allergy test for CBD formulation. Patients and members of the study group (dentists, who performed and collected the results of muscle activity using surface electromyography (sEMG) were blinded for allocation and treatment.

The inclusion criteria were as follows:Patients who agree to participate in the studyPatient’s age must be within ≥18 and ≤60 yearsGood general healthTemporomandibular disorder–positive as per the Polish version of the Research Diagnostic Criteria for Temporomandibular Disorders (RDC/TMD) for group Ia and Ib [30].Presence of all teeth (with the exception of third molars)

The exclusion criteria were as follows:Cannabis formulation/placebo formulation allergyHypersensitivity to substances to be used in the studySkin wounds with skin surface discontinuationAddiction to cannabisPatients being treated with analgesic drugs and/or drugs that affect muscle function Fixed or removable dental prosthesis Disease or autoimmune disorder associated with generalized muscular tension

This study was approved by the Bioethical Committee of the Silesian Medical Chamber in Katowice, Poland (number KNW/0022/KB1/7/19) and is retrospectively registered at ClinicalTrials.gov (NCT03994640). This study was performed in accordance with the Declaration of Helsinki and the International Conference on Harmonization: Guidelines for Good Clinical Practice. Patients received verbal and written information describing the trial and gave their consent to participate in the study.

### 2.2. Study Protocol

This study followed the consolidated standards of reporting trials (CONSORT) statement and was performed between 01.01.2018 and 01.01.2019 in the Department of Temporomandibular Disorders at the Medical University of Silesia in Katowice, Poland.

The trial consisted of two main visits, preceded by screening:(1)Screening visit for study participation and inclusion.(2)Baseline visit for sEMG test (baseline EMG I, VAS I) and random allocation to the group, receiving a packaging with formulation (CBD or placebo) along with the instructions on how to use it.(3)First follow-up visit after 14 days of application (follow-up EMG II, VAS II).

Patients were examined by experienced dentists (AN, KW). Randomization was performed by ANB, a dentist, who was not involved in follow-up visits. The sEMG tests were performed by KWD on Day 0 and Day 14. KWD used the same paper templates to place electrodes in exactly the same position on follow-up visits on Day 0 and Day 14, to achieve comparable results. She was not aware of the patient’s group.

### 2.3. sEMG Measurements

Masseter muscle tension (μV) at rest and in maximal contraction was measured in previously determined points, with four-channel Neurobit Optima 4.0 (Neurobit Systems, Poland). Points were repeatably marked on patients’ skin on baseline visit (EMG I) and on follow-up visit (EMG II) in the region of masseter with a special paper template. Two dischargeable electrodes with a diameter of 10 mm were placed with conductor gel on each of the masseter muscle: near the origin, under the zygomatic arch, and on the angle of the mandible, approximately 10 mm from each other. The location of the electrodes was based on anatomical landmarks, palpated by the dental practitioner during the examination of stomatognathic system. Patients were asked to perform an isometric contraction of the masseter muscles to find the best place to fix the electrode.

The placement of the electrodes, as well as preparation of the skin, was consistent with SENIAM guidelines: facial hair was shaved if necessary and cleaned with alcohol (Surface ElectroMyoGraphy for the Non-Invasive Assessment of Muscles, www.seniam.org). Four Ag/AgCl adhesive electrodes, diameter 30 mm (Sorimex, Toruń, Poland), were applied bilaterally, directly on both the masseter muscles (under the zygomatic arch and on the angle of the mandible) with an interelectrode distance of 10 mm. Patients remained in a silent and calm room and were made to sit in an upright position with feet on the floor and looking forward. The occlusal plane was parallel to the floor. Reference electrodes were placed on the neck. The sEMG signal was amplified and digitized.

First, sEMG signals of the resting position (RP) of the mandible were recorded. It was done without the teeth contact and the position of the mandible was maintained only by the force of gravity and viscoelasticity of the stomatognathic system tissues. Patients were asked to slightly open the mouth until the teeth of the upper and lower jaw were out of contact. Then, the sEMG records of RP from right and left masseter muscles were recorded during a maximal voluntary isometric contraction (MVIC) and was established as the reference value: the tests were repeated thrice for each muscle on each side. The tests were separated by at least 1 min of rest. The mean values of sEMG (μV) of the masticatory muscles in mandible’s RP and in MVIC, obtained from Bioexplorer Neurobit (Neurobit Optima for sEMG, Neurobit Systems, Gdynia, Poland were analyzed and normalized. Results were analyzed with Statistica 13.1 (Stat Soft, Krakow, Poland).

### 2.4. Normalization of sEMG Values

Normalization of the data collected allows for the assessment of the level of activity of the masseter muscle during RP as compared to the maximal activation capacity of the masseter muscle [31,32]. The data were normalized by dividing the values by the values of the MVIC. Reliability of the normalized data was higher, compared to nonnormalized data [33,34].

### 2.5. Topical Application of CBD or Placebo Formulation

Patients in Group1 were given CBD formulation, which was prepared according to the individual recipes for testing purposes. Many different vehicles were tested in order to obtain the best features of the CBD formulation. Cannabis oil and Charlotte’s Web Hemp Extract Oil Formula Olive Oil 30 mL (A00674, product code 910.061 Stanley Brothers Boulder CO80301, cannabinoid content is given in Table 2). In Hemp Extract Oil, the concentration of CBD was 7.3% (66.97 mg CBD/mL, 0.461 mg CBD-A/mL, and 0.28 mg CBD-V/mL). The density of Hemp Extract Oil was 925 mg/mL. Assumptions of maximum concentration and lightweight formulation were adopted. The formula was developed on the basis of ointments, which is characterized by optimal water binding properties and ease of oil-water emulsion formation.

#### 2.5.1. Testing of Vehicle

With cholesterol as the vehicle, experimental ointment (for Group1) with 20% CBD oil was prepared. The approximate content of CBD (CBD, CBD-A, and CBD-V) in this formulation was 1.46%.

The composition of cholesterol ointment (per 100 g) according to Polish Pharmacopoeia XI, on the basis of which a formulation with CBD oil was prepared:Cholesterol 3.0 gSolid paraffin 15.0 gLiquid paraffin 64.0 gWhite vaseline 18.0 g

It was decided to develop a formulation based on cholesterol ointment due to the lack of pharmaceutical incompatibility with the test hemp oil containing CBD. Cholesterol ointment is a type of official pharmacopoeial medium used in pharmacy as an absorption medium, i.e., an emulsifying active substance dissolved in a solvent, which is most often water. Cholesterol ointment can be used to make various formulations as a vehicle, it can be also an independent preparation in an unprocessed form. A large “amount of water”, defined as the amount of water in grams that 100 g of substrate permanently binds at 20 °C of 120 for the substrate used, allows to emulsify a large amount of water and aqueous solutions. The presence of water in the formulation will reduce the feeling of greasiness—the preparation will have a lighter consistency and will allow the introduction of any additional active substances or water-soluble preservatives into the drug. The discussed medium can be applied to thermally and mechanically damaged skin, in atopic dermatitis and after long-term treatment with steroid ointments, so any pathological changes at the application site will not be a restriction on the use of the preparation. All ingredients are safe for the skin and do not cause irritation. There are also no specific contraindications for the use of this ointment base.

#### 2.5.2. Preparation of CBD Formulation

Oleum CBD 2.0 g (20% CBD oil)Aqua purificata 3.0 gUng. Cholesterol 5.0 g

Patients in Group2 were given control formulation, based on cholesterol as the vehicle but without CBD oil. The following is the composition of control formulation:

#### 2.5.3. Preparation of Control Formulation

Aqua purificata 3.0 gUng. Cholesterol 5.0 g

The ointments had the same consistency and color, they were packed in the same containers, which made it impossible to distinguish the formulation with CBD oil from placebo.

In both groups, the formulation was intended for topical use to be applied on the skin surface of the masseter muscle, at the right and left side. Containers with CBD formulation were marked with even and odd digits for the appropriate group to allocate patient in one of the two groups. In addition to patients, doctors did not know the identity of the formulation. First, patients were asked to perform an allergy test by applying a small amount of the formulation (placebo or CBD) on the skin of the forearm 24 h before therapy. In the case of swelling, itching, or redness on skin, the patients were asked to quit therapy.

Each patient had been taught on the procedure to apply the formulation in equal amounts (the size of peas) on both sides. Patients were informed that the formulation should be applied and rubbed gently into the skin surface (approximately 4 × 4 cm) and were supposed to apply it twice a day for up to a period of 14 days before the follow-up visit.

### 2.6. Sample Size Estimation

Values of the minimum sample size were determined below in the case of Student’s *t*-tests for dependent samples, assuming the target values of test power equal to 95% (0.95) and 99% (0.99) and the significance level of the test α = 0.05. The values were generated using the STATISTICA software package, version 13.1. The results are presented in the table below (Table 3):

### 2.7. Statistical Analysis of Masseter Muscles Electromyographic Activity 

The data obtained in experimental Group1 and in control Group 2 are available in the form of dependent samples (the value of the statistical feature is verified before and after the placebo/CBD formulation application), with an interval of 14 days. In statistical analysis, to demonstrate the effectiveness of CBD formulation, the following statistical tests for dependent samples were used:Student’s *t*-test for dependent samplesWilcoxon signed-rank testSign test

#### 2.7.1. Student’s *t*-test for Dependent Samples

According to the Student’s *t*-test, since the group sizes were rather small (*n* = 30), we performed the Shapiro–Wilk normality test. Table 4 presents the results of Shapiro–Wilk normality test.

Table 5 shows the results of Student’s *t*-test. The significance level of 0.05 was assumed.

The null hypothesis $H_{0}$ is the normality of the distribution of the appropriate variable, with an assumed significance level of *α* = 0.05.

#### 2.7.2. Wilcoxon Signed-Rank Tests

Table 6 presents the results of a Wilcoxon signed-rank test (compared to the values of appropriate variables on days 0 and 14). A significance level of 0.05 was assumed.

We obtained results similar to that of Student’s *t*-test. For the patients treated with CBD formulation, there was a significant improvement in masseter sEMG activity (the decrease of average rest masseter muscle activity values).

#### 2.7.3. Sign Test

An alternative to the Wilcoxon signed-rank test is the Sign test. Sign test provided the following results in Table 7 (assumed for a significance level 0.05):

As it was shown in the sign test, a significant decrease in electromyographic activity of masseter muscles was noted in patients treated with CBD formulation. The results of sign test match Wilcoxon signed-rank test results. Table 8 presents the normalized mean and the standard deviation of the level of activity in rest position (RP).

#### 2.7.4. Comparison of the Two Independent Groups (Control Formulation and CBD Formulation)

Based on the results of Student’s *t*-test and Mann–Whitney *U*-test, the effect of the treatment with CBD formulation is clearly visible. In the case of Wald–Wolfowitz test, there is no basis to reject the null hypothesis; however, in the case of comparison of p-values, the significance level is close to 0.05.

Table 9 shows the results (*p*-values) of the statistical analysis.

### 2.8. Statistical Analysis of Pain Intensity Changes

#### 2.8.1. *t*-Student Tests for Dependent Samples

We verify the null hypothesis: $m_{1}=m_{2},$ where $m_{i}$ stands for the mean value of the considered variable in the *i*th population, i=1, 2. The results of the *t*-student tests are given in Table 10. The significance level 0.05 is assumed.

As one can observe, for the Group1 of patients treated with CBD formulation the difference between means on Day0 and Day14 is very large and the null hypothesis will be rejected at any level of significance (*p*-value is approximately 0). In the case of Group2, patients treated with placebo, however the hypothesis is rejected for significance level 0.05.

#### 2.8.2. Wilcoxon Signed-Ranks Tests

To verify the null hypothesis that the use CBD formulation has no significant impact, we will use the Wilcoxon signed-ranks test. The results of the test (we compare the values of appropriate variables on Day0 and on Day14) presents Table 11. We assume significance level 0.05.

A conclusion similar to that formulated for t-Student test can be obtained. For the patients treated with CBD formulation in Group1, the zero hypothesis will be rejected at any level of significance. In the case of the Group2 (patients treated with placebo) the result depends on the level of significance. Such a situation shows the essential influence of the treatment with CBD formulation for the average pain level (the treatment gives an essential reduction of the pain level).

#### 2.8.3. Sign Test

The sign test gives the following results (the decision is taken for significance level 0.05): Table 12 shows the results (*p*-values) of the statistical analysis.

As one can observe from the sign test follows a significant impact of the treatment CBD formulation in Group1 for the decreasing of the average pain level (the conclusion the same as in the case of the Wilcoxon signed-ranks test).

In Table 13 the results for the main descriptive statistics are presented, namely the mean and the standard deviation for all variables in two groups.

#### 2.8.4. Comparison of Two Independent Groups (Group1 and Group2)

In comparing the results between Group1 and Group2 after 14 days of therapy, we use the following tests:Student’s *t*-test for independent samples means (the null hypothesis is that means are the same in two Groups)Wald-Wolfowitz runs test and U Mann- Whitney test (the null hypothesis is that probability distributions are the same in Group1 and Group2)

The results (*p*-values) are shown in Table 14:

As one can observe, if we take any significance level (not necessary 0.05), the null hypothesis will be rejected according to all considered tests, so the impact of the treatment with CBD formulation in Group1 is visibly essential. 

## 3. Results

### 3.1. Masseter Muscles sEMG Activity Changes

According to our results, there were no statistically significant differences between the two groups in terms of age or gender (*p* > 0.05). In the case of Group1 after 14 day, there was a significant effect of CBD on the average level of the rest sEMG activity of masseter muscles. In the case of Group2 after 14 days, there was no effect of using the formulation on the average level of masseter muscle sEMG activity.

The sEMG masseter muscle activity was significantly decreased in Group1 (CBD formulation) (11% in the right masseter muscle and 12.6% in the left masseter muscle) compared to Group2 (placebo formulation) (0.23% in the right masseter muscle and 3.3% in the left masseter muscle). The reduction in the mean values of sEMG after normalization of rest masseter was statistically significant in Group1 and was not significant in Group2 (Figure 4).

### 3.2. Pain Intensity Changes in VAS Scale

In the case of Group1, there was a significant effect of CBD on the average pain level of masseter muscles. In the case of Group2, there was no effect of using the formulation on the average pain level of masseter muscle. It was observed in the study that pain intensity in VAS scale after 14 days of application of CBD formulation over masseter muscles was significantly decreased in Group1 (CBD) from the average VASI = 5.6 (SD = 1.38) on Day 0, to VAII = 1.67 (SD = 1.44). Comparing to Group2: placebo formulation application from the average VASI = 5.10 (SD = 1.26) on Day 0 to VASII = 4.60 (SD = 1.58) on Day 14. The pain intensity in VAS scale was significantly decreased in Group1 (CBD formulation): 70.2% reduction comparing to Group2 (placebo formulation): 9.81% reduction. The reduction was statistically significant in Group1 and was not significant in Group2 (Figure 5).

### 3.3. Adverse Effects

There were no adverse effects recorded in this study.

## 4. Discussion

CBD is a non-psychoactive substance that can affect the treatment of many disease states, including pain and inflammation [10], which is confirmed by in vivo studies showing CBD activity to reduce the release of proinflammatory cytokines [35]. The multitude of potential uses of cannabinoids causes a change in their perception, from recreational herbal compounds to specific drugs used in many civilization diseases among which we can distinguish MFP.

Inhalations are the most popular way of administering the active substances contained in cannabis [36]. The systemic bioavailability of CBD administered by this route is estimated at 31% [36]. The bioavailability of cannabidiol and delta-9-tetrahydrocannabinol in an aerosol prescribed for oral use is estimated at 6%, and the delay of action is 120 min, much longer compared to the previously mentioned inhalation [36]. Due to the limited oral bioavailability [37] caused by digestion, the specificity of the location of MFP and the lipophilic nature of the test compound, it was decided to test transdermal CBD administration [1,4]. Transdermal drug formulations have many advantages over oral forms of drugs. Transdermal administration avoids the first-pass metabolism effect and thus improves the bioavailability of the drug. In addition, transdermal administration allows for constant release of the drug for a longer period of time at the application site, while minimizing the adverse effects of higher drug concentrations, which in turn can improve the efficacy and safety of the patient’s pharmacotherapy. The topical application of medicinal substances is ideal for inflammation and muscle pain.

Transdermal cannabinoids are effective in reducing pain and inflammatory symptoms [38,39]. Potential disadvantages of the percutaneous solution used, manifesting as a possible local allergic reaction or low penetration of the drug with a hydrophilic structure can be quickly refuted. Both the carrier in the form of cholesterol ointment and the active substance in the form of CBD are hypoallergenic and have a lyophilic character, significantly facilitating penetration through the skin [4,40]. A very interesting idea that may be useful in the future in myofascial pain therapy is transdermal administration through the patches as a CBD carrier, providing even greater control over the process of release of the active substance [37].

According to Lucas, CBD transdermal administration is 10 times higher than administration of THC in humans [37]. Among all cannabinoids CBD with the most polar chemical structure should be in favor for improved skin absorption [37]. That is why authors decided to evaluate CBD cream efficiency in reducing superficial electromyographic activity of masseters muscles in patients with myofascial pain. Cannabinoids are accumulated in adipose tissues and well vascularized organs. Cannabidiol has a long elimination half-life: 24 h after intravenous administration and 2–5 days after repeated, daily inhalation, respectively [37].

Cannabinoids are used, inter alia, in treatment of nausea and vomiting in patients after chemotherapy [10] to increase appetite in patients with AIDS, anorexia nervosa [11], and the pain associated with multiple sclerosis, epilepsy (particularly in children), neurodegenerative diseases—such as Parkinson’s disease or Huntington’s disease. In addition, THC has antimicrobial properties—it inhibits the growth of gram-positive bacteria [4].

Wong has found that intramuscular injection of THC reversed the mechanical sensitization, induced in masseter muscle of rats [25]. Authors found cannabinoids CB1 and CB2 receptors in trigeminal ganglion, that innervated rats masseter muscles. Results of the research study on rats suggest that cannabinoid receptors are a target for analgesic therapy in muscle pain disorders. Because of the difficulty in administering CBD to the site of action in the CB1 and CB2 receptor region, transdermal administration seems to be optimal. In this way, the alimentary canal and intramuscular injections are omitted. Cannabidiol, one of many cannabinoids in cannabis sativa, can be delivered through the skin surface. The level of transdermally absorbed CBD was measured in blood plasma in rat-model research [36]. After 0.62 mg CBD oil application on 3.5 cm^2^ skin surface, 3.8 ng/mL (SD = 1.4 ng/mL) CBD was found in blood plasma [36]. CBD formulation, tested in this research was approximately 1.46% CBD, compared to 10% CBD gel used in rat-model of the transdermal cannabidiol application [4]. In this research study, the application was local, over the masseter muscle, so the authors are of the opinion that local plasma levels might be lower compared with concentrated 10% gel [1]. Further research is needed to determine the appropriate dose and method of CBD administration.

According to Stinchcomb expected therapeutic plasma level of CBD is 10 ng/mL in therapy of nausea vomiting, but also in pain therapy [41]. The strategy “start low and go slow” seems to be the best solution in optimal cannabinoids prescribing, choosing the best dose for a patient [37]. The ideal method of transdermal use of CBD would be the aforementioned transdermal patches, with the active substance in the form of CBD, worn on painful muscles for a long time.

No psycho-active side effects were observed in patients attending control visits, because there was no THC in the cannabis oil, and there were no cases of hypothermia or hypomobility induced by THC.

The surface electromyography (sEMG) recordings were performed at mandibular rest position from the masseter region and results were compared intra-session and inter-sessions after 7 and 14 days. A frequent complaint against studies comparing EMG values is the lack of reproducibility of sEMG measurements. However, according to a study published by Castroflorio in 2005, the analysis of the masseter muscle activity was considered repeatable, both between visits and during one study [42]. Authors of the study have observed that electrode placement is critical for low-level EMG activity analysis, that is why they have proposed the use of the template for sEMG follow-up visits [42]. The template was also used during this research study.

Dysregulation of the balance between sympathetic and parasympathetic part within the autonomous nervous system is probably a very important factor in the induction of bruxism and myofascial pain. The autonomic nervous system regulates the body’s unconscious actions, for example sleep bruxism. Innovative methods to reduce muscle tonus and allow masticatory muscles to recover form myofascial pain are in demand [43]. CBD as a safe alternative to THC should be carefully studied in all possible applications in medicine. Advanced clinical trials should be conducted to determine the dose, route of administration and effectiveness of CBD in the therapy of myofascial pain of masseter muscles. CBD is a nonpsychoactive substance with its possible use in pain therapy, in multiple sclerosis and in the treatment of inflammation [10]. The oral bioavailability of CBD is limited because of the action of digestive enzymes, but in vivo study has demonstrated its activity in reducing the processes of inflammation and cytokine release [36]. Aerosol with CBD and delta-9-THC is prescribed for use in the oral cavity as an anti-inflammatory therapy [37]. The oral bioavailability of both THC and CBD is poor which has been evaluated to be approximately 6% and the time to action is 120 min, which is much longer when compared to cannabis inhalation [38].

## 5. Conclusions

The application of CBD formulation on the masseter muscle reduces sEMG activity and pain intensity of masseter muscles during RP and improves the condition of the muscle in patients with MFP. Further research is needed in this field, but CBD, as an alternative for THC, should be taken into consideration in the therapy of masticatory muscles in patients with TMD.

## Figures and Tables

**Figure 1 jcm-08-01886-f001:**
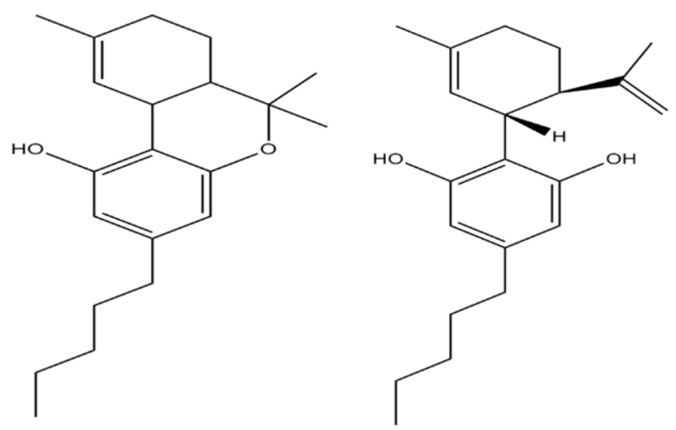
2D ∆9-tetrahydrocannabinol structure and cannabidiol structure.

**Figure 2 jcm-08-01886-f002:**
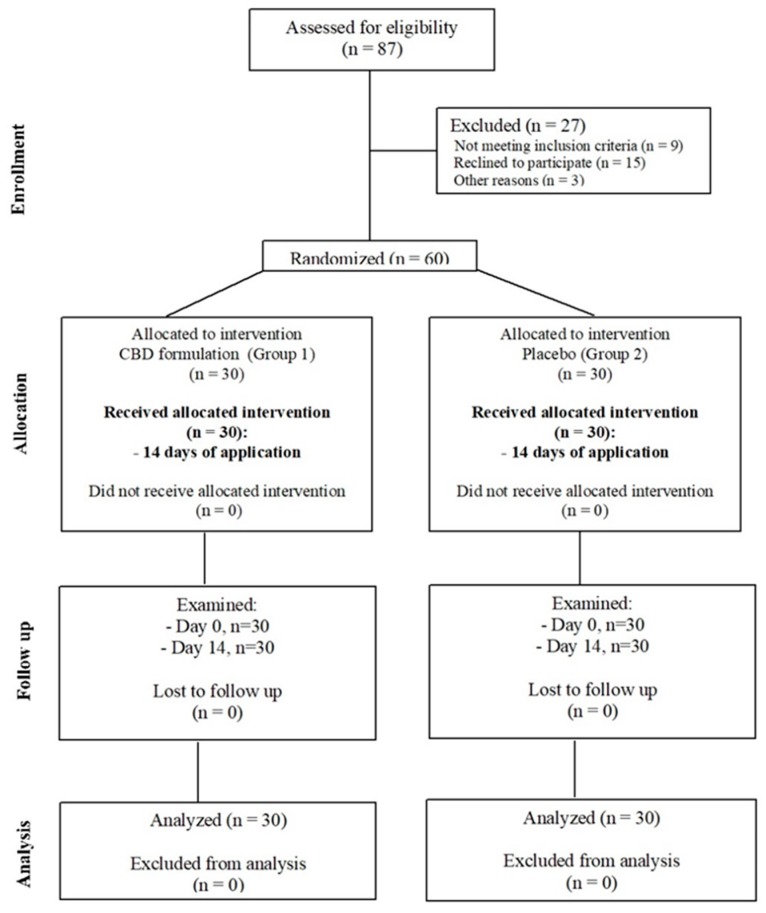
Flow chart of the two arms consolidated standards of reporting trials (CONSORT)-randomized study.

**Figure 3 jcm-08-01886-f003:**
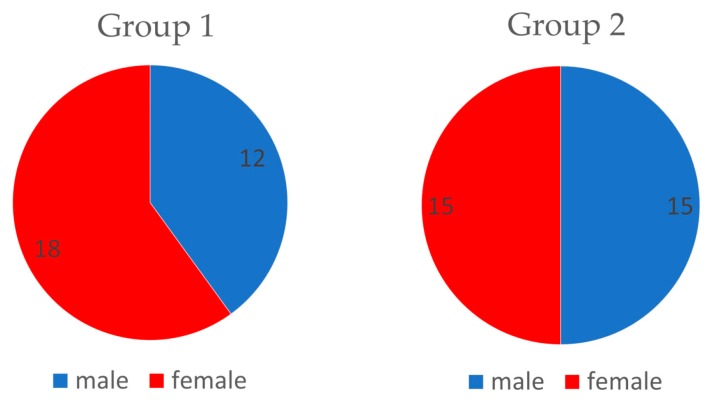
Distribution of gender in experimental and control groups.

**Figure 4 jcm-08-01886-f004:**
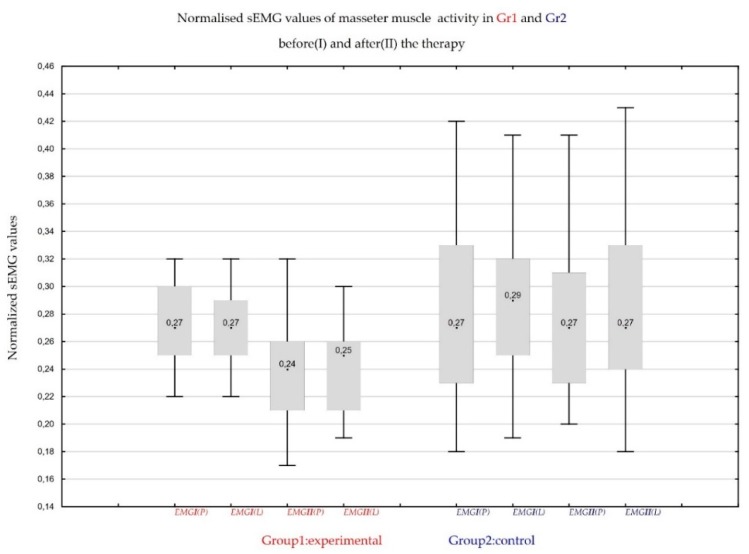
Normalized surface electromyographic (sEMG) mean values of masseter muscle activity at rest position in Group1 and Group2; on Day 0 (I) and on follow-up visit on Day 14 (II) of the therapy; on left (L) and right (R) masseter muscle.

**Figure 5 jcm-08-01886-f005:**
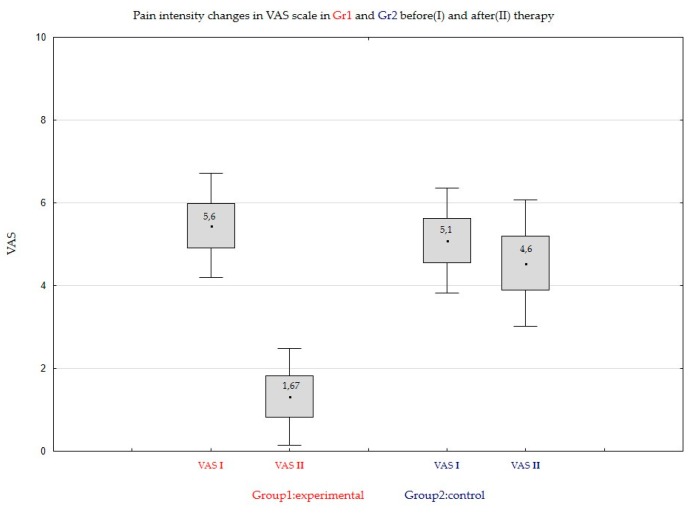
Pain intensity changes in VAS scale in Group1 and Group2 on Day0 (I) and on Day 14 (II) of the therapy.

**Table 1 jcm-08-01886-t001:** Structure of Group I and Group II main descriptive values.

Variables		Group1	Group2
*N*	60	30	30
Sex	Female/male	18/12	15/15
Age		23.2 (SD = 1.6)	22.6 (SD = 1.86)
RDC/TMD	Ia	9	13
	Ib	21	17

**Table 2 jcm-08-01886-t002:** Cannabinoid content in Charlotte’s Web Hemp Extract Oil Formula Olive Oil (according to Stanley Brothers Boulder Certificate).

Parameter	Concentration
Hemp Extract	70 mg/serving
Cannabinoids	
THC	0.00 mg/mL
THC-A	<0.0467 mg/mL
THC-V	<0.0467 mg/mL
CBD	66.97 mg/mL
CBD-A	0.461 mg/mL
CBD-V	0.280 mg/mL
CBG	2.05 mg/mL
CBG-A	<0.0467 mg/mL
CBN	<0.0467 mg/mL
CBC	3.74 mg/mL

**Table 3 jcm-08-01886-t003:** Minimal sample size for Student’s *t*-tests for dependent samples (TP = test power).

Group	Variable	Correlation	Minimal Sample Size (TP = 0.95)	Minimal Sample Size (TP = 0.99)
1	EMG I(P) vs. EMG II(P)	0.49282210	17 (TP = 0.9563)	23 (TP = 0.9915)
1	EMG I(L) vs. EMG II(L)	0.39039193	15 (TP = 0.9502)	21 (TP = 0.9919)
2	EMG I(P) vs. EMG II(P)	0.77346774	64047 (TP = 0.9500)	90551 (TP = 0.9900)
2	EMG I(L) vs. EMG II(L)	0.46762877	404 (TP = 0.9502)	570 (TP = 0.9900)

**Table 4 jcm-08-01886-t004:** Normality tests of all variables.

Group	Variable	Test Statistic *W*	*p*-Value
1	EMG I(P)	0.96100	0.32840
1	EMG I(L)	0.96796	0.48497
1	EMG II(P)	0.94409	0.11721
1	EMG II(L)	0.96608	0.43811
2	EMG I(P)	0.94729	0.14295
2	EMG I(L)	0.95238	0.19573
2	EMG II(P)	0.93287	0.05855
2	EMG II(L)	0.96379	0.38573

**Table 5 jcm-08-01886-t005:** Student’s *t*-test for means of dependent samples.

Variable	Sample Difference in Means	Test Statistic *T*	*p*-Value
Group1(P)	0.036333	5.198612	0.000015
Group1(L)	0.034000	5.493967	0.000006
Group2(P)	0.000667	0.077980	0.938379
Group2(L)	0.012333	0.985221	0.332664

**Table 6 jcm-08-01886-t006:** Results of Wilcoxon signed-rank tests.

Variable	Test Statistic *W*	*p*-Value
Group1(P)	5.500000	0.000005
Group1(L)	5.500000	0.000005
Group2(P)	178.00000	0.569163
Group2(L)	194.00000	0.611352

**Table 7 jcm-08-01886-t007:** Results of sign tests.

Variable	*p*-Value
Group1(P)	0.000008
Group1(L)	0.000001
Group2(P)	0.850107
Group2(L)	0.710347

**Table 8 jcm-08-01886-t008:** Main descriptive statistics.

Group	Variable	Mean	Standard Deviation
1	EMG I(P)	0.273333	0.026824
1	EMG I(L)	0.270667	0.026514
1	EMG II(P)	0.237000	0.043561
1	EMG II(L)	0.236667	0.033869
2	EMG I(P)	0.292000	0.070925
2	EMG I(L)	0.299000	0.068247
2	EMG II(P)	0.291333	0.067962
2	EMG II(L)	0.286667	0.064505

**Table 9 jcm-08-01886-t009:** Comparison of Group1 “cannabidiol (CBD)” and Group2 “placebo” group.

Variable	Student *t*-Test	Wald–Wolfowitz Runs Test	Mann–Whitney *U*-test
Group1/Group2(P)	0.005000	0.434659	0.002157
Group1/Group2(L)	0.000000	0.068318	0.001638

**Table 10 jcm-08-01886-t010:** *t*-Student tests for means of dependent samples.

Variable	Sample Difference Means	Test Statistic *T*	*p*-Value
Group1 VAS	3.93	13.31971	0.000000
Group2 VAS	0.50	2.715305	0.011038

**Table 11 jcm-08-01886-t011:** Results of Wilcoxon signed-ranks tests.

Variable	Test Statistic W	*p*-Value
Group1 VAS	0.00	0.000003
Group2 VAS	37.50	0.020672

**Table 12 jcm-08-01886-t012:** Results of sign tests.

Variable	*p*-Value
Group1 VAS	0.000000
Group2 VAS	0.021781

**Table 13 jcm-08-01886-t013:** Main descriptive statistics for pain intensity changes in VAS Visual Analogue Scale.

Variable	Mean	Standard Deviation
VASI Group1	5.60	1.379655
VASII Group1	1.67	1.446359
VASI Group2	5.10	1.268994
VASII Group2	4.60	1.588754

**Table 14 jcm-08-01886-t014:** Comparison of Group1 and Group2 after 14 days.

Variable	*t*-Student Test	Wald-Wolfowitz Runs Test	*U* Mann-Whitney Test
VASII Group1/VASII Group2	0.000000	0.000159	0.000000

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
