# Peer review of "Myorelaxant Effect of Transdermal Cannabidiol Application in Patients with TMD: A Randomized, Double-Blind Trial"

_jcm, 2019, doi:10.3390/jcm8111886_

Round 1

Reviewer 1 Report

I reviewed your manuscript titled:”Myorelaxant effect of transdermal cannabidiol application in patients with TMD: a randomized, double blind trial” and found it relevant to the study of pain, and as such of interest to the readers of the Journal of Clinical Medicine

Nevertheless, some minor points of criticisms should be addressed before the manuscript can be considered for publication.

My comments are the following:

The first paragraph of the introduction have to be rephrase according to the expert consensus published in 2018 (Lobbezoo F, Ahlberg J, Raphael KG, et al. International consensus on the assessment of bruxism: report of a work in progress. J Oral Rehabil. 2018: 45:837-844.) A paragraph regarding MFP (diagnosis, aetiology, management) is needed

Authors are required to explain why they refrain from performing pain evaluation at the beginning of the study and during the cream application. This is in my opinion a study limitation since MFP is basically a pain disorder

Author Response

Thank You for Your review of the manuscript and for all You valuable comments.

These are the answers for Your questions:

Point.1. The first paragraph of the introduction have to be rephrased according to the expert consensus published in 2018(Lobbezzo F, Ahlberg J, Raphael KG et al. International consensus on the assesment of bruxism: report of a work in progress. J Oral Rehabil. 2018:45:837-844.) 

Answer.1.First paragraph of the introduction has been rephrased according to International consensus on the assessment of bruxism 2018.

Point.2.A paragraph regarding MFP( diagnosis, aetiology, management) is needed.

Answer.2. A paragraph regarding myofascial pain aetiology, diagnosis and management was added into the manuscript.

Point.3. Authors are required to explain why they refrain from performing pain evaluation at the beginning of the study and during the cream application. this is in my opinion a study limitation since MFP is basically a pain disorder.

Answer.3. Pain assessment was carried out and was one of the mail goals of our research work, unfortunately an error crept in placing the final version of the manuscript in the Journal of Clinical Medicine System, for which we are very sorry. All pain analysis data are included in the text and highlighted in yellow.

Regarding moderate english changes we have sent the manuscript before submission to Journal of Clinical Medicine to TRANSLMED for editing. We are waiting for another correction and the certificate of English Correction after checking some minor changes in reviewed manuscript. 

Thank You for Your patience.

Best regards Aleksandra Nitecka-Buchta  

Reviewer 2 Report

Introduction section needs to be improved in order to achieve a better background. 

Methods section , please add how sample 

Size was obtained. 

Please remove decision about hypothesis in tables

Semien Criteria was used in EMG procedure? 

Please use anova in statistics

Discussion section seems Confused, please rewrite

Author Response

Thank You for Your review of the manuscript and all You valuable comments.

These are answers for Your questions:

Point.1. Introduction section needs to be improved in order to achieve a better background.

Answer.1. A paragraph regarding myofascial pain was added to the introduction and highlighted in yellow.

Point.2.Methods section, please add how sample size was obtained.

Answer.2. The issue of the minimum sample size occurs in the case of interval estimation(confidence intervals), when we want to determine how many elements the sample should consist of in order to ensure the realization of the confidence interval(e.g. for the mean value) with a sufficiently high probability (usually 95% or 99%). This issue is also being considered in the case of statistical tests. Then, for a determined level ∝ of significance of the test, the minimum sample size needed to guarantee a sufficiently high test power is determined (also usually at the level of 95% or 99%). Values of the minimum sample size were determined below in the case of Student's t-tests for dependent samples, assuming the target values of the test power equal to 95%(0.95) and 99%(0.99) and the significance level of the test  ∝=0.05. The values were generated using the STATISTICA software package, version 13.1. The results are presented in the table.

Point.3. Please remove decision about hypothesis in tables

Aswer.3. The column with decision information was removed from the tables.

Point.4.Semien Criteria was used in EMG procedure?

Answer.4. SENIAM (Surface ElectroMyoGraphy for the Non-Invasive Assessment of Muscles) project guidelines are recommended by  Biomedical Health and Research Program (BIOMED II) of the European Union. The project resulted in recommendations for sensors and sensors placement procedures and signal processing methods for sEMG. Recommendations were introduced to standardize the research methodology of sEMG research.

Point.5. Please use ANOVA in statistics

Answer.5. ANOVA(Analysis of Variance) is a statistical technique used, among others to verify the hypothesis about the equality of means in the case of k populations, where k>2. Because in our study there are two comparative groups: experimental and control ones(k=2), classic ANOVA analysis cannot be used here, although the test of equality of two means can be called a special case of ANOVA.

Point.6. Discussion section seems confused please rewrite

Answer.6. Discussion section has been rewritten and reorganised.

Regarding moderate english changes, we have sent the manuscript before submission to Journal of Clinical Medicine to TRANSLMED for editing. We are waiting now for another correction and the Certificate of English Correction after checking some minor changes in reviewed manuscript. 

Thank You for Your patience

Best regards Aleksandra Nitecka-Buchta
